# Birth-related retinal hemorrhages: The Soonchunhyang University Cheonan Hospital universal newborn eye screening (SUCH-NES) study

**In Hwan Cho[1], Min Seong Kim[1], Nam Hun Heo[2], So Young Kim[1] ***

**1** Department of Ophthalmology, College of Medicine, Soonchunhyang University, Cheonan, Korea,
**2** Clinical Trial Center, Soonchunhyang University, Cheonan, Korea

* ophdrkim@gmail.com

## Abstract

### Purpose

To report the prevalence, related factors, and characteristics of birth-related retinal hemorrhages (RHs) according to their severity in healthy newborns using a telemedicine network and wide-field digital retinal imaging (WFDRI).

### Methods

Newborns who underwent WFDRI at 61 obstetrics/gynecology hospitals between January 2017 and December 2019 were enrolled. Demographics and related factors were compared among newborns with and without RHs. The newborns' eyes were divided into the minimal, mild, moderate, and severe groups according to the number of RHs, and characteristics like bilaterality, laterality, involved retinal layer, involved zone, macular and/or optic nerve (ON) involvement were compared.

### Results

Among 56247 newborns, 13026 had birth-related RHs (23.2%). Normal spontaneous vaginal delivery (NSVD) showed the highest association with RHs (odds ratio, 19.774; 95% confidence interval, 18.277–21.393; $P < 0.001$) on multivariate analysis. Bilateral RHs (8414/13026; 64.59%) were more common than unilateral RHs (4612/13026; 35.41%); however, unilateral RHs (2383/4217; 56.51%) were more common than bilateral RHs (1834/4217; 43.49%) in the minimal group. RHs showed no laterality differences between the two eyes ($P = 0.493$). Most RHs were intraretinal (18678/21440; 87.12%), and 2328 (31.65%) eyes with preretinal hemorrhage were observed in the severe group. Zone I RHs were common in the minimal (7072/7090; 99.75%), mild (4953/4960; 99.86%), and moderate (2013/2035; 98.92%) groups; zone I and II RHs were common in the severe group (4843/7355; 65.85%); and RHs in zone III were rare (7/21440; 0.03%). Most RHs showed no macular and/or ON involvement in the minimal and mild group; however, this was common in the severe group (7111/7355; 96.68%).

**Data Availability Statement:** All data files are available from https://dataverse.harvard.edu/dataset.xhtml?persistentId=doi:10.7910/DVN/7SS3A6.

**Funding:** So Young Kim is supported by
SoonChunHyang Research Fund. The funders had
no role in study design, data collection and
analysis, decision to publish, or preparation of the
manuscript.

**Competing interests:** The authors have declared
that no competing interests exist.

## Conclusions

Birth-related RHs were common in healthy newborns and were significantly associated with
NSVD. RHs were usually bilateral, intraretinal, and distributed posterior to the retina, but
severe RHs had unique characteristics. Future long-term and longitudinal studies are
required to elucidate the prognosis of severe RHs.

## Introduction

Birth-related retinal hemorrhages (RHs) are known to occur in healthy newborns [1–12]. The
characteristics of these RHs have been reported by many researchers, but these vary widely
and are inconclusive [1–12]. A study by Emerson et al. [11] reported that most RHs were
intraretinal and primarily involved the posterior retina. However, a study by Callaway et al. [5]
reported that most RHs were multilayered and involved the peripheral retina. These discrepan-
cies may be due to the wide clinical spectrum of birth-related RHs, differences in the age at
examination, and varying examination methods used in each study [5, 8, 9, 11–13]. Therefore,
a study with a larger number of newborns undergoing objective examination at a similar age
would be required to verify the results. Furthermore, the characteristics of birth-related RHs
seem to depend on their severity. Severe RHs may persist for a longer duration and potentially
result in visual disturbances, such as anisometropia and amblyopia, while mild RHs spontane-
ously disappear without clinical significance [3, 8, 9, 11, 12, 14]. However, only a few studies
with limited sample sizes have reported the characteristics of RHs according to their severity,
and a generalized agreement has not yet been established [3, 8, 9, 11]. Investigating the charac-
teristics of RHs according to severity will also be meaningful.

Telemedicine combined with wide-field digital retinal imaging (WFDRI) may be an effec-
tive and feasible method for screening a large number of newborns with RHs at a similar
examination age [3, 5, 15–17]. This strategy involves capturing retinal fundus images and
transmitting the remote interpretations of pediatric ophthalmologists [17]. The images were
captured by trained nursing staff as soon after birth as possible, and the interpretations were
performed by analyzing the photographs. Therefore, the process was less labor- and time-
intensive than conventional binocular indirect ophthalmoscopy (BIO) examination, which
necessitated direct contact between the patients and clinicians [14]. Furthermore, more accu-
rate and objective documentation of RHs is possible, because WFDRI has a wider field of view
than conventional BIO and can provide digital records [3, 17].

The Soonchunhyang University Cheonan Hospital universal newborn eye screening
(SUCH-NES) study is an institutional prospective cohort study that used the telemedicine sys-
tem to determine the prevalence, characteristics of ocular problems, and the long-term visual
outcomes of newborns with ocular abnormalities in healthy newborns. In this study, we
enrolled newborns from the SUCH-NES study and investigated the prevalence, related factors,
and characteristics of RHs according to their severity.

## Materials and methods

### Subjects

Newborns who were born at 61 obstetrics/gynecology (OB/GYN) hospitals (Fig 1) between
January 2017 and December 2019 were asked to participate in this study. Since fundus photog-
raphy was optional, only newborns whose parents gave written consent to the examination of

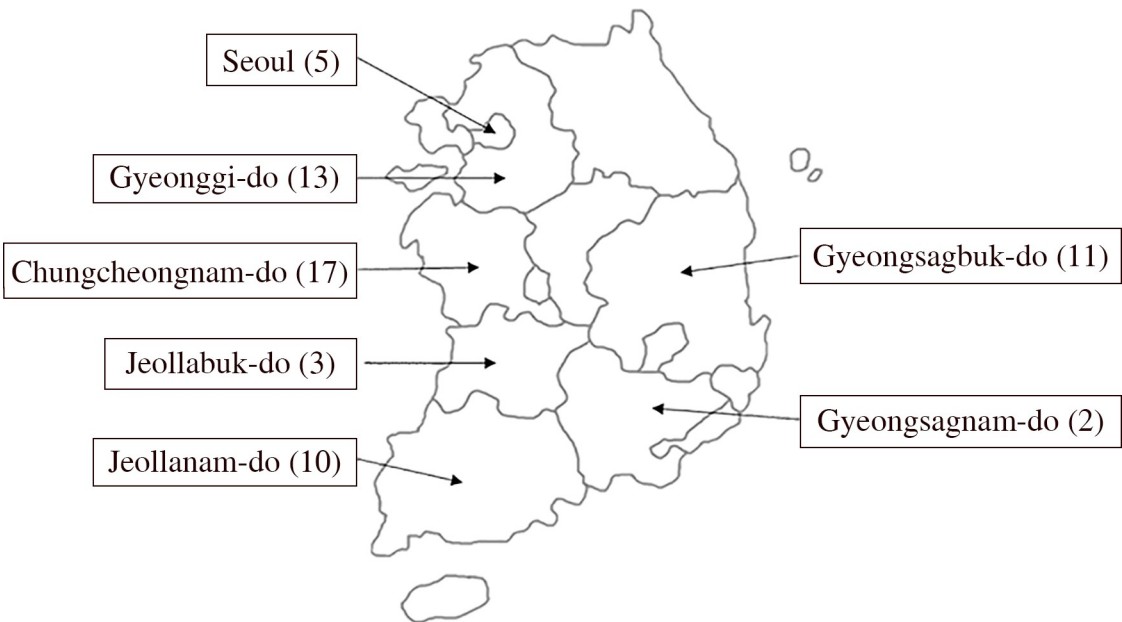

**Fig 1. Distribution of the participating obstetrics/gynecology hospitals.** The numbers in parentheses indicate the number of hospitals.

their child were screened. Newborns with (1) gestational age (GA) less than 32 weeks and/or birth weight (BW) less than 2000 g, (2) maternal and fetal systemic diseases, (3) structural ocular anomalies, (4) a familial history of ocular congenital anomalies, and (5) those transferred to tertiary hospitals due to problems during delivery or (6) those undergoing the examination 7 days after birth were excluded from the study. Sex, mode of delivery (normal spontaneous vaginal delivery [NSVD] or cesarean section [CS]), GA, BW, and Apgar scores (1 and 5 min) of each newborn were recorded. The Institutional Review Board (2019-04-007) of Soonchunhyang University approved the study, and it adhered to the tenets of the Declaration of Helsinki.

## Photography protocols

External eye and retinal images were captured by adequately trained nursing staff at each OB/GYN hospitals using WFDRI (RetCam III; Clarity Medical Systems, Pleasanton, CA). The photography protocol was as follows: (1) dilatation of the pupil with a mixture of 0.5% tropicamide and 0.5% phenylephrine eye drops; (2) imaging of the eyelid and anterior segment of both the eyes; (3) topical anesthesia using 0.5% proparacaine hydrochloride eye drops; (4) insertion of an eye speculum to open the eye; (5) imaging of red reflex assessment using illumination from RetCam III (Clarity Medical Systems); and (6) imaging of the entire retina using five-directional photography with an inert lubricating jelly and RetCam 130˚ lens. The five-directional fundus photographs included the posterior pole, optic nerve (ON) centered, ON superior, ON inferior, and ON nasal views. All examinations were performed with close cardiac and respiratory monitoring. Topical antibiotic eye drops were administered at the end of the examination.

## Telemedicine network

At the end of the working hours, the images of newborns that were captured at the 61 OB/GYN hospitals were uploaded to a virtual hard drive, which was accessible using laptops and

smartphones. These images were then interpreted by one pediatric vitreoretinal specialist (SYK) within 24 h. If the captured images were of poor quality, the newborns were re-screened. Since the newborns were hospitalized for 48 h after NSVD and 72 h after CS, an effort was made to report the results of the newborn eye examination to the parents within 48 hours. If abnormal ocular findings were noticed, the newborns were referred to a nearby pediatric ophthalmologist immediately. In the case of newborns with severe RHs, monthly follow-up was recommended until the RHs disappeared.

## Analysis of birth-related RHs

Retinal images from newborns with RHs were reviewed by two independent graders (SYK and IHC) in a masked fashion. If there was any discrepancy between the graders, consensus was reached by discussion. The following factors were collected and analyzed: bilateralities of RHs, lateralities of RHs, numbers of RHs, size of the largest RH, involved retinal layers, involved zone, involvement of the macula and/or ON, involvement of the fovea, accompanying vitreous hemorrhage (VH), and presence of Roth spots. As per the modified classification by Watts et al, newborns and eyes with RHs were divided into four groups according to the number of RHs affecting the eyes. Less than 10 RHs were defined as minimal, 10 to 30 RHs as mild, 30 to 50 RHs as moderate, and more than 50 RHs as severe (Fig 2). If the severity of RHs in both the eyes did not match, the groups of newborns were classified on the basis of the more severe eye [8].

The zone of involvement was defined according to zones I, II, and III of retinopathy of prematurity [18]. Zone I was defined as the circle, the radius of which is twice the distance between the center of the optic disc and the center of the macula. Zone II extended centrifugally from the edge of zone I to the nasal ora serrata. Zone III was the residual crescent of the retina anterior to zone II. In addition, the severity of RHs was scored using a grading scale similar to that described by Binenbaum et al. (Table 1) [13, 19].

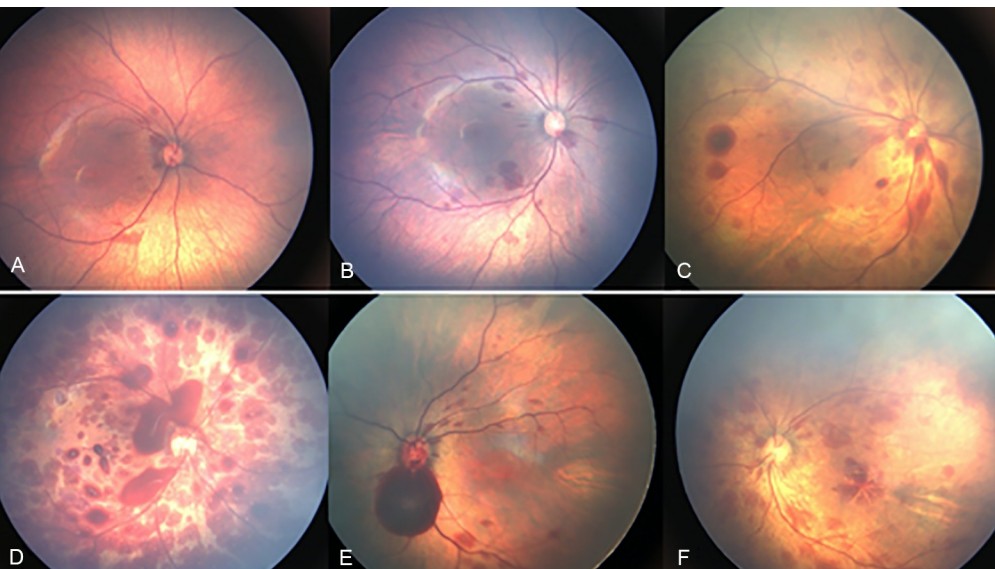

**Fig 2. Representative fundus photographs of newborns with retinal hemorrhages.** (A) Minimal retinal hemorrhage. (B) Mild retinal hemorrhage. (C) Moderate retinal hemorrhages. (D) Severe retinal hemorrhages with vitreous hemorrhage. (E) Optic nerve involvement of the hemorrhages. (F) Macular involvement of the hemorrhages.

**Table 1. Grading scales of retinal hemorrhages[a].**

| Retinal hemorrhage | Points |
|---|---|
| **Type and size (only one category chosen)** | |
| **Mild:** Intraretinal hemorrhage only | 1 |
| **Moderate:** Subhyaloid hemorrhage present; all lesions less than two disc areas in size | 2 |
| **Severe:** Subhyaloid hemorrhage; vitreous hemorrhage or any lesion greater than two disc areas in size | 3 |
| **Extent (sum of categories)** | |
| **Any hemorrhage within the following areas** | |
| Macula (>1 DD from the disc; within 2 DD of the fovea) | 1 |
| Peripapillary (within 2 DD of the disc, excluding the macula) | 1 |
| Periphery (outside the above region) | 1 |

DD: disc diameter.

[a]The overall score is calculated by combining the total scores for each eye; the maximal score is 6 points per eye or 12 points overall.

## Statistical analysis

The mean and standard deviation of each variable were calculated using IBM SPSS Statistics for Windows, Version 20.0 (IBM Corporation, Armonk, NY). Student's t-test or analysis of variance was used for comparing continuous variables, and the chi-squared test was used for categorical variables. Scheffe's post-hoc test was used to analyze the statistical differences among the groups. The factors related to RHs in newborns were assessed using univariate linear regression analysis. Variables selected for the univariate analysis were sex, BW, GA, Apgar scores at 1 and 5 min, delivery method, and age at examination. Predictors with a $P$ less than 0.05 in the univariate analysis were entered into the multivariate regression analysis. The agreement of the severity between eyes was calculated using weighted kappa. A $P$ less than 0.05 was considered to indicate statistical significance.

## Results

### Newborns with and without birth-related RHs

Among the screened newborns, newborns who met the inclusion criteria were enrolled. In total, 56247 newborns from 61 OB/GYN hospitals were examined over 3 years. Overall demographics of these newborns are described in Table 2.

Demographics were compared between newborns with and without birth-related RHs. Among the 56247 newborns, 13026 had RHs, and the prevalence was 23.2%. A significant difference existed in the mode of delivery between the two groups ($P < 0.001$). Newborns without RHs were mostly delivered via CS (24684/43221; 57.11%) rather than NSVD (18537/43221; 42.89%). However, newborns with RHs were more frequently delivered via NSVD (12289/13026; 94.34%) rather than CS (737/13026; 5.66%). Notably, 39.87% (12289/30826) of newborns delivered via NSVD developed RHs, whereas only 2.90% (737/25421) of newborns delivered via CS developed RHs. BW (3233.04 ± 414.32 vs 3234.70 ± 373.02 g; $P = 0.665$) was not significantly different between the two groups. However, GA (272.69 ± 8.40 vs 274.47 ± 7.64 days; $P < 0.001$) was significantly longer, Apgar scores at 1 minute (8.94 ± 0.66 vs 8.98 ± 0.62; $P < 0.001$) and 5 minutes (9.79 ± 0.47 vs 9.82 ± 0.42; $P < 0.001$) were significantly higher, and age at examination (2.18 ± 2.04 vs 1.38 ± 1.16; $P < 0.001$) was significantly lower in newborns with RHs than in those without RHs (Table 2).

**Table 2. Demographics of newborns with and without retinal hemorrhages.**

|  | Total | Newborns without RHs | Newborns with RHs | *P*-value |
|---|---|---|---|---|
| N (%) | 56247 (100.0) | 43221 (76.8) | 13026 (23.2) |  |
| Sex, n (%) |  |  |  |  |
| 0: Male | 28894 (51.37) | 22361 (51.74) | 6533 (50.15) | 0.002[a] |
| 1: Female | 27353 (48.63) | 20860 (48.26) | 6493 (49.85) |  |
| Delivery method, n (%) |  |  |  |  |
| NSVD | 30826 (54.80) | 18537 (42.89) | 12289 (94.34) | <0.001[a] |
| CS | 25421 (45.20) | 24684 (57.11) | 737 (5.66) |  |
| Gestational age (days), mean ± SD | 273.10 ± 8.27 | 272.69 ± 8.40 | 274.47 ± 7.64 | <0.001[b] |
| Birth weight (g), mean ± SD | 3233.42 ± 405.13 | 3233.04 ± 414.32 | 3234.70 ± 373.02 | 0.665[b] |
| Agar score (1 min), mean ± SD | 8.95 ± 0.65 | 8.94 ± 0.66 | 8.98 ± 0.62 | <0.001[b] |
| Agar score (5 min), mean ± SD | 9.79 ± 0.46 | 9.79 ± 0.47 | 9.82 ± 0.42 | <0.001[b] |
| Age at examination (days), mean ± SD | 2.00 ± 1.90 | 2.18 ± 2.04 | 1.38 ± 1.16 | <0.001[b] |

CS: Cesarean section; NSVD: normal spontaneous vaginal delivery; RHs: retinal hemorrhages; SD: standard deviation.

[a]Chi-square test.

[b]Student's t-test.

## Factors related to birth-related RHs in newborns

Univariate and multivariate analyses were performed to identify the predictors of birth-related RHs in newborns (Table 3). The univariate linear regression analysis showed that the following parameters were significantly associated with RHs: female sex (odds ratio [OR], 1.065; 95% confidence interval [CI], 1.024–1.108; $P < 0.001$), NSVD (OR, 22.204; 95% CI, 20.564–23.974; $P < 0.001$), GA (OR, 1.028; 95% CI, 1.025–1.030; $P < 0.001$), Apgar score at 1 minute (OR, 1.125; 95% CI, 1.091–1.160; $P < 0.001$), Apgar score at 5 min (OR, 1.194; 95% CI, 1.141–1.249; $P < 0.001$), and age at examination (OR, 0.631; 95% CI, 0.619–0.644; $P < 0.001$). Similarly, the multivariate analysis revealed that female sex (OR, 1.050; 95% CI, 1.005–1.098; $P = 0.029$), NSVD (OR, 19.774; 95% CI, 18.277–21.393; $P < 0.001$), and age at examination (OR, 0.797; 95% CI, 0.782–0.812; $P < 0.001$) was significantly associated with RHs (Table 3).

**Table 3. Univariate and multivariate analyses of factors related to retinal hemorrhages.**

|  | Univariate analysis | | Multivariate analysis | |
|---|---|---|---|---|
|  | OR (95% CI) | *P*-value | OR (95% CI) | *P*-value |
| Sex, n (%) |  |  |  |  |
| Male | 1 |  | 1 |  |
| Female | 1.065 (1.024–1.108) | 0.002 | 1.050 (1.005–1.098) | 0.029 |
| Delivery method |  |  |  |  |
| NSVD | 22.204 (20.564–23.974) | <0.001 | 19.774 (18.277–21.393) | <0.001 |
| CS | 1 |  | 1 |  |
| Birth weight (g) | 1.000 (1.000–1.000) | 0.627 |  |  |
| Gestational age (days) | 1.028 (1.025–1.030) | <0.001 | 0.994 (0.992–0.997) | 0.241 |
| Apgar score (1 min) | 1.125 (1.091–1.160) | <0.001 | 1.018 (0.963–1.076) | 0.541 |
| Apgar score (5 min) | 1.194 (1.141–1.249) | <0.001 | 1.078 (0.996–1.168) | 0.063 |
| Age at examination (days) | 0.631 (0.619–0.644) | <0.001 | 0.797 (0.782–0.812) | <0.001 |

CS: Cesarean section; CI: confidence interval; NSVD: normal spontaneous vaginal delivery; OR: odds ratio.

**Table 4. Demographics of newborns according to the severity of retinal hemorrhages.**

| | Total | Minimal | Mild | Moderate | Severe | *P*-value |
|---|---|---|---|---|---|---|
| Newborns, n (%) | 13026 (100.00) | 4217 (32.37) | 3002 (23.05) | 1348 (10.35) | 4459 (34.23) | |
| Bilateral RHs, n (%) | 8414 (64.59) | 1834 (43.49) | 1728 (57.56) | 925 (68.62) | 3927 (88.07) | <0.001[a] |
| Unilateral RHs, n (%) | 4612 (35.41) | 2383 (56.51) | 1274 (42.44) | 423 (31.38) | 532 (11.93) | |
| Sex, n (%) | | | | | | |
| Male | 6537 (50.18) | 2109 (50.01) | 1531 (51.00) | 695 (51.56) | 2202 (49.38) | 0.358 |
| Female | 6489 (49.82) | 2108 (49.99) | 1471 (49.00) | 653 (48.44) | 2257 (50.62) | |
| Delivery method | | | | | | |
| NSVD | 12289 (94.34) | 3863 (91.61) | 2826 (94.14) | 1263 (93.69) | 4337 (97.26) | <0.001[b] |
| CS | 737 (5.66) | 354 (8.39) | 176 (5.86) | 85 (6.31) | 122 (2.74) | |
| Gestational age (days), mean ± SD | 274.47 ± 7.64 | 274.56 ± 7.76 | 274.06 ± 7.79 | 274.10 ± 7.45 | 274.76 ± 7.44 | 0.471 |
| Birth weight (g), mean ± SD | 3234.70 ± 373.02 | 3235.33 ± 372.75 | 3233.38 ± 374.25 | 3239.88 ± 371.70 | 3232.06 ± 373.09 | 0.916 |
| Agar score (1 min), mean ± SD | 8.98 ± 0.62 | 8.97 ± 0.62 | 8.99 ± 0.63 | 8.96 ± 0.61 | 9.00 ± 0.62 | 0.058 |
| Agar score (5 min), mean ± SD | 9.82 ± 0.42 | 9.82 ± 0.42 | 9.82 ± 0.42 | 9.82 ± 0.43 | 9.83 ± 0.40 | 0.632 |
| Age at examination (days), mean ± SD | 1.38 ± 1.16 | 1.50 ± 1.38 | 1.41 ± 1.14 | 1.37 ± 1.27 | 1.24 ± 0.85 | <0.001[c] |

CS: Cesarean section; NSVD: normal spontaneous vaginal delivery; RHs: retinal hemorrhages; SD: standard deviation.

[a]Bilateral RHs vs unilateral RHs.

[b]NSVD vs CS.

[c]The severe group is significantly different from the minimal ($P < 0.001$), mild ($P < 0.001$), and moderate ($P = 0.002$) groups (Scheffe's post-hoc analysis).

## Demographics of newborns according to the severity of RHs

Newborns with birth-related RHs (13026; 100%) were divided into the minimal (4217; 32.37%), mild (3002; 23.05%), moderate (1348; 10.35%), and severe (4459; 34.23%) groups according to the number of RHs. Overall, bilateral RHs (8414/13,026; 64.59%) were more common than unilateral RHs (4612/13026; 35.41%). However, a difference in bilaterality was observed according to the severity of RHs. Bilateral RHs were more common than unilateral RHs in the mild (1728/3002; 57.56% vs 1274/3002; 42.44%), moderate (925/1348; 68.62% vs 423/1348; 31.38%), and severe (3927/4459; 88.07% vs 532/4459; 11.93%) groups. In the minimal group, unilateral RHs (2383/4217; 56.51%) were more common than bilateral RHs (1834/4217; 43.49%) (Table 4). The severity of bilateral RHs was usually consistent between the eyes, and the percentage of agreement was 71.4% ($P < 0.01$) (Table 5).

No significant differences existed among the groups in terms of the sex ratio ($P = 0.358$), GA ($P = 0.471$), BW ($P = 0.916$), Apgar score at 1 min ($P = 0.058$), and Apgar score at 5 min ($P = 0.632$). The mode of delivery ($P < 0.001$) and age at examination ($P < 0.001$) were significantly different among the groups. NSVD was more common than CS in the minimal (3863/4217; 91.61% vs 354/4217, 8.39%), mild (2826/3002; 94.14% vs 176/3002; 5.86%), moderate (1263/1348; 93.69% vs 85/1348; 6.31%), and severe (4337/4459; 97.26% vs 122/4459; 2.74%) groups. In the post-hoc analysis, age at examination of the severe group (1.24 ± 0.85 days) was

**Table 5. Agreement of the inter-eye severity of retinal hemorrhages.**

| Left eye \ Right eye | Minimal | Mild | Moderate | Severe | Total |
|---|---|---|---|---|---|
| **Minimal** | **1834 (80.09)** | 144 (7.89) | 133 (16.16) | 306 (8.80) | 2417 (28.73) |
| **Mild** | 134 (5.85) | **1450 (79.50)** | 21 (2.55) | 257 (7.39) | 1862 (22.13) |
| **Moderate** | 104 (4.54) | 15 (0.82) | **652 (79.22)** | 18 (0.52) | 789 (9.38) |
| **Severe** | 218 (9.52) | 215 (11.79) | 17 (2.07) | **2896 (83.29)** | 3346 (39.77) |
| **Total** | 2290 (100.00) | 1824 (100.00) | 823 (100.00) | 3477 (100.00) | **8414 (100.00)** |

significantly lower than that of the minimal (1.50 ± 1.38; $P < 0.001$), mild (1.41 ± 1.14; $P < 0.001$), and moderate (1.37 ± 1.27; $P = 0.002$) groups (Table 4).

## Characteristics of eyes with RHs according to their severity

Since 8414 newborns had bilateral birth-related RHs, 21440 newborn eyes were included in the analysis of characteristics according to the severity of RHs. Newborn eyes with RHs were also divided into the minimal (7090; 33.07%), mild (4960; 23.13%), moderate (2035; 9.49%), and severe (7355; 34.31%) groups according to the number of RHs. There was no laterality difference in RHs between the eyes, and it was not significantly different among the groups ($P = 0.493$) (Table 6).

A significant difference was observed among the groups in terms of the number of RHs, size of the largest spot, and RH grading score (all $P < 0.001$). In the post-hoc analysis, the

**Table 6. Characteristics of eyes with retinal hemorrhages according to severity.**

|  | Total | Minimal | Mild | Moderate | Severe | P-value |
|---|---|---|---|---|---|---|
| Eyes with RHs, n (%) | 21440 (100.00) | 7090 (33.07) | 4960 (23.13) | 2035 (9.49) | 7355 (34.31) |  |
| Right eye, n (%) | 10871 (50.70) | 3571 (50.37) | 2508 (50.56) | 1013 (49.78) | 3779 (51.38) | 0.493[a] |
| Left eye, n (%) | 10569 (49.30) | 3519 (49.63) | 2452 (49.44) | 1022 (50.22) | 3576 (48.62) |  |
| Number of RHs, n (%) | 14.75 ± 12.97 | 4.75 ± 2.38 | 18.41 ± 5.00 | 40.68 ± 5.46 | >50 | <0.001[b] |
| Size of the largest spot, DD (%), mean ± SD | 0.79 ± 0.93 | 0.22 ± 0.19 | 0.39 ± 0.29 | 0.56 ± 0.35 | 1.66 ± 1.10 | <0.001[c] |
| RH grading score (each eye), mean ± SD | 2.30 ± 1.61 | 1.05 ± 0.24 | 1.34 ± 0.53 | 1.94 ± 0.59 | 4.25 ± 1.11 | <0.001[d] |
| Involved layer, n (%) |  |  |  |  |  |  |
| Preretinal | 2463 (11.49) | 37 (0.52) | 58 (1.17) | 40 (1.97) | 2328 (31.65) | <0.001 |
| Intraretinal | 18678 (87.12) | 7043 (99.34) | 4893 (98.65) | 1990 (97.79) | 4752 (64.61) |  |
| Subretinal | 19 (0.09) | 0 (0.00) | 1 (0.02) | 2 (0.10) | 16 (0.22) |  |
| All layers | 280 (1.31) | 10 (0.14) | 8 (0.16) | 3 (0.15) | 259 (3.52) |  |
| Involved zone, n (%) |  |  |  |  |  |  |
| Zone I | 16545 (77.17) | 7072 (99.75) | 4953 (99.86) | 2013 (98.92) | 2507 (34.09) | <0.001 |
| Zone II | 28 (0.13) | 15 (0.21) | 5 (0.10) | 7 (0.34) | 1 (0.01) |  |
| Zones I and II | 4860 (22.67) | 3 (0.04) | 2 (0.04) | 12 (0.59) | 4843 (65.85) |  |
| Zone III | 7 (0.03) | 0 (0.00) | 0 (0.00) | 3 (0.10) | 4 (0.05) |  |
| Macular and/or ON involvement, n (%) |  |  |  |  |  |  |
| No involvement | 10558 (49.24) | 6770 (95.49) | 3380 (68.15) | 401 (19.71) | 7 (0.10) | <0.001 |
| Macula only | 530 (2.47) | 40 (0.56) | 207 (4.17) | 239 (11.74) | 44 (0.60) |  |
| ON only | 2978 (13.89) | 278 (3.92) | 1318 (26.57) | 1189 (58.43) | 193 (2.62) |  |
| Macula and ON | 7374 (34.39) | 2 (0.03) | 55 (1.11) | 206 (10.12) | 7111 (96.68) |  |
| Foveal involvement, n (%) |  |  |  |  |  |  |
| Yes | 416 (1.94) | 8 (0.11) | 11 (0.22) | 23 (1.13) | 374 (5.08) | <0.001 |
| No | 21024 (98.06) | 7082 (99.89) | 4949 (99.78) | 2012 (98.87) | 6981 (94.92) |  |
| Vitreous hemorrhage, n (%) |  |  |  |  |  |  |
| Yes | 285 (1.33) | 11 (0.16) | 10 (0.20) | 8 (0.39) | 256 (3.48) | <0.001 |
| No | 21,155 (98.67) | 7,079 (99.84) | 4,950 (99.80) | 2,027 (99.61) | 7,099 (96.52) |  |
| Roth spot |  |  |  |  |  |  |
| Yes | 10319 (48.13) | 466 (6.57) | 1491 (30.06) | 1299 (63.83) | 7063 (96.03) | <0.001 |
| No | 11121 (51.87) | 6624 (93.43) | 3469 (69.94) | 736 (36.17) | 292 (3.97) |  |

DD: Disc diameter; ON: optic nerve; RHs: retinal hemorrhages; SD: standard deviation.

[a]: Right eyes vs left eyes.

[b]: The severe group is significantly different from the minimal ($P < 0.001$), mild ($P < 0.001$), and moderate ($P < 0.001$) groups (Scheffe's post-hoc analysis).

[c]: The severe group is significantly different from the minimal ($P < 0.001$), mild ($P < 0.001$), and moderate ($P < 0.001$) groups (Scheffe's post-hoc analysis).

[d]: The severe group is significantly different from the minimal ($P < 0.001$), mild ($P < 0.001$), and moderate ($P < 0.001$) groups (Scheffe's post-hoc analysis).

number of RHs (>50) in the severe group was significantly higher than that in the minimal (4.75 ± 2.38; $P < 0.001$), mild (18.41 ± 5.00; $P < 0.001$), and moderate (40.68 ± 5.46; $P < 0.001$) groups. The size of the largest spot (1.66 ± 1.10 disc diameter) in the severe group was significantly greater than that in the minimal (0.22 ± 0.19; $P < 0.001$), mild (0.39 ± 0.29; $P < 0.001$), and moderate (0.56 ± 0.35; $P < 0.001$) groups. RH grading score (4.25 ± 1.10) in the severe group was significantly higher than that in the minimal (1.05 ± 0.24; $P < 0.001$), mild (1.34 ± 0.53; $P < 0.001$), and moderate (1.94 ± 0.59; $P < 0.001$) groups (Table 6).

The prevalence distribution of the involved layer, involved zone, involvement of the macula and/or ON, involvement of the fovea, accompanying VH, and presence of Roth spots was significantly different among the groups (all $P < 0.001$). Overall, most RHs were intraretinal hemorrhages (18678/21440; 87.12%), including those in the minimal (7043/7090; 99.34%), mild (4893/4960; 98.65%), moderate (1990/2035; 97.79%), and severe (4752/7355; 64.61%) groups. Notably, in the severe group, 2328 eyes (31.65%) showed preretinal hemorrhages, and RHs involved all the retinal layers in 259 eyes (3.52%). RHs were mainly distributed only in zone I (16545/21440; 77.17%), including those in the minimal (7072/7090; 99.75%), mild (4953/4960; 99.86%), and moderate (2013/2035; 98.92%) groups. However, RHs were mainly distributed in zones I and II in the severe group (4843/7355; 65.85%). RHs solely in zone III were rarely found, and included only 3 eyes (0.10%) in the moderate group and 4 eyes (0.05%) in the severe group. Most RHs did not involve the ON or macula in the minimal (6770/7090; 95.49%) and mild (3380/4960; 68.15%) groups. However, only ON involvement showed a high percentage in the moderate (1189/2035; 58.43%) group, and both macular and ON involvement showed a high percentage in the severe (7111/7355; 96.68%) group. Most of the RHs did not involve the fovea, but 374 (5.08%) eyes with severe RHs showed foveal involvement. Accompanying VH (256/7355; 3.48%) and Roth spots (7063/7355; 96.03%) were most commonly detected in the severe group (Table 6). Atypical features of RHs, such as subhyaloid hemorrhages with tortuous retinal vessels, RHs with vessel anomaly, VH with ON involvement, and sectoral RHs, were also observed (Fig 3).

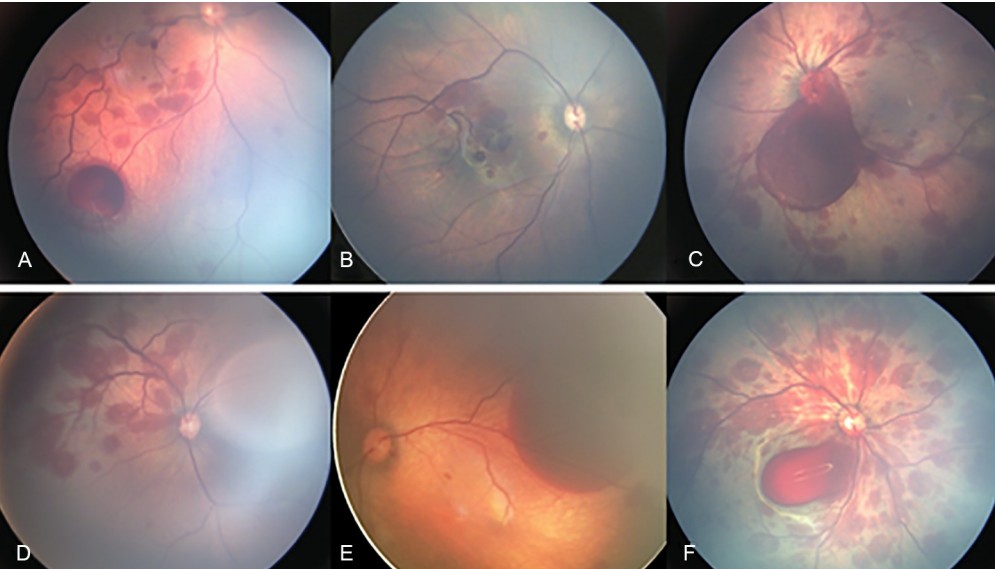

**Fig 3. Atypical features of newborns with retinal hemorrhages.** (A) Subhyaloid hemorrhages with tortuous retinal vessels. (B) Retinal hemorrhages with vessel anomalies. (C) Vitreous hemorrhage with optic nerve involvement. (D) Sectoral retinal hemorrhages. (E) Subretinal hemorrhages. (F) Dense vitreous hemorrhage with macular involvement.

### Follow-up of severe RHs

Among the 7355 eyes with severe RHs, 2859 (38.87%) eyes were re-examined monthly after birth at our ophthalmology department. Of these, most of the RHs disappeared; however, 126 (4.40%) eyes showed persistent RHs until 2 months after birth (Fig 4).

## Discussion

In the present study, we successfully screened a large number of newborns from 61 OB/GYN hospitals using a telemedicine network combined with WFDRI. To the best of our knowledge, this is the largest study investigating the characteristics of newborns with RHs. In our study, the prevalence of birth-related RHs was 23.2%. This value is comparable with those reported in previous studies by Chen et al. (22%) [3], Li et al. (19% and 21.52%, respectively) [4, 9], Callaway et al. (20.3%) [5], and Zhao et al. (24.5%) [6]. However, it is higher than those reported by Ma et al. (6.7%) [1], Goyal et al. (13.2%) [2], and Vinekar et al. (2.4%) [7]. These

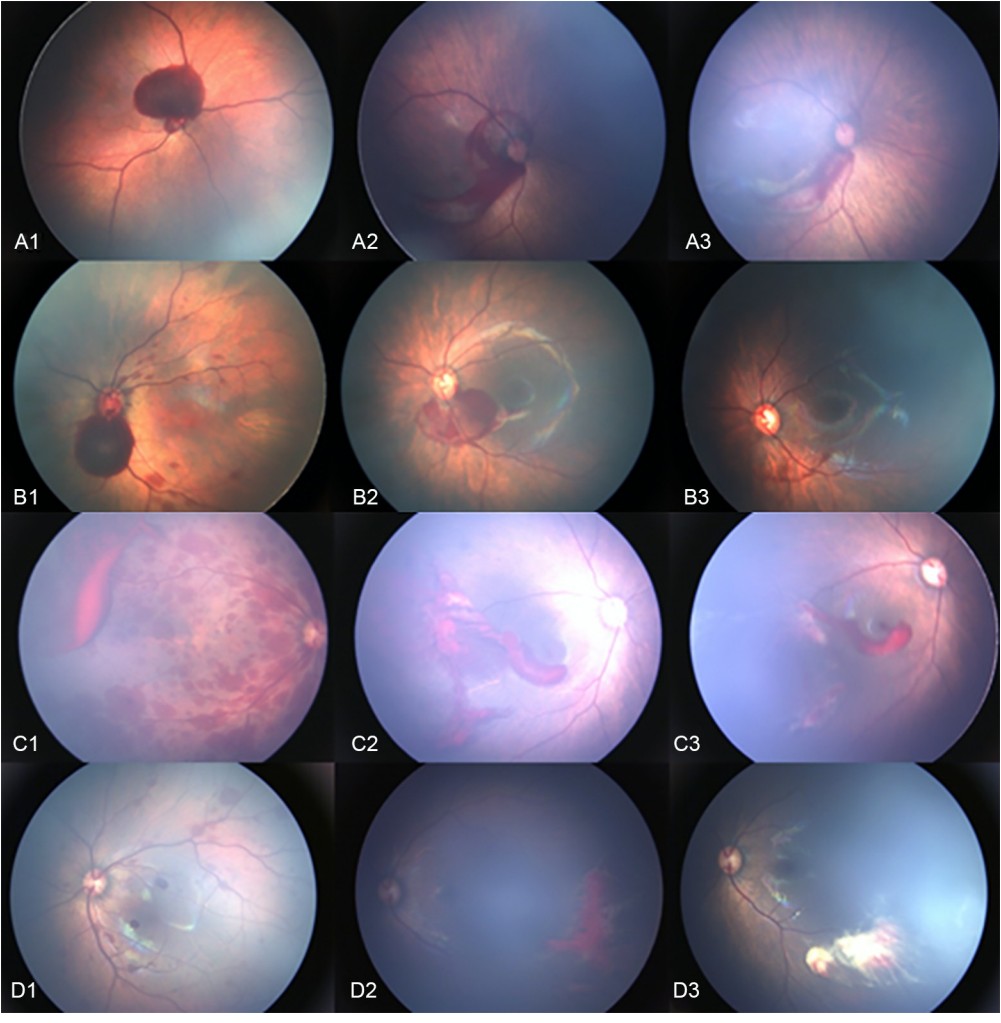

**Fig 4. Representative fundus photographs of eyes with persistent retinal hemorrhages during follow-up.** (A1-D1) Fundus photographs of RHs at birth. (A2-D2) Photographs from the follow-up examination at 1 month after birth. (A3-D3) Photographs from the follow-up examination at 2 months after birth. RH: retinal hemorrhages.

discrepancies may be due to the differences in the age at examination [1, 2] and demographics of the newborns [7].

Birth-related RHs seem to occur because of the compression of the head during passage through the birth canal. An acute increase in intracranial pressure due to compression inhibits central retinal venous flow and subsequently results in an acute change in central retinal arterial pressure [5, 12, 13]. In other words, the retina of newborns becomes more vulnerable to hemorrhages during vaginal delivery. In our study, 94.34% of newborns with RHs were delivered via NSVD, and only 5.66% were delivered via CS. While 39.87% of newborns delivered via NSVD developed RHs, only 2.90% of newborns delivered via CS developed RHs. Furthermore, multivariate analysis confirmed that NSVD significantly increased the odds of RHs in newborns (OR, 19.774; 95% CI, 18.277–21.393; $P < 0.001$). These results are comparable with previous findings that the mode of delivery, including NSVD, forceps-assisted vaginal delivery, and vacuum-assisted vaginal delivery, are significantly associated with the occurrence of RHs in newborns [2, 3, 5, 6, 8, 10, 11, 20, 21]. A study by Goyal et al. [2] reported RHs in 47.6% of newborns delivered via NSVD but only 5.2% of those delivered via CS. A study by Callaway et al. [5] reported that NSVD significantly increased the odds of fundus hemorrhages than did CS (OR, 9.34; 95% CI, 2.5–33.97). A study by Zhao et al. [6] reported that NSVD was positively correlated (OR, 3.81; 95% CI, 2.65–5.48) and CS was negatively correlated (OR, 0.30; 95% CI, 0.14–0.63) with the occurrence of RHs. A study by Emerson et al. [11] reported that 75% of newborns delivered via vacuum-assisted vaginal delivery developed RHs, and a study by Williams et al. [21] reported that vacuum- and forceps-assisted vaginal deliveries were associated with the occurrence of severe RHs. Unfortunately, since none of the newborns included in our study were delivered via vacuum- or forceps-assisted vaginal delivery, we could not elucidate the differences in the occurrence of RHs according to the mode of delivery.

Other demographic factors, including sex ratio, GA, Apgar scores at 1 and 5 min, and age at examination, were significantly different between the newborns with or without RHs. However, as the magnitude of differences of GA and Apgar scores at 1 and 5 min were minimal between the newborns with and without RHs, the clinical significance of these factors might be ambiguous.

In the multivariate analysis, GA, BW, and Apgar scores were not significantly associated with the occurrence of RHs, and these results were consistent with those of previous studies [6, 11]. Female sex (OR, 1.050; 95% CI, 1.005–1.098; $P = 0.029$) was significantly associated with RHs. However, the OR was slightly higher than 1, and the clinical significance of female sex prevalence compared to male sex prevalence was ambiguous. Further studies are warranted to validate these results. Age at examination (OR, 0.797; 95% CI, 0.782–0.812; $P < 0.001$) was also significantly associated with RHs, and this was in agreement with the findings of previous studies [22, 23]. A study by Giles et al. [22] reported that the occurrence of RHs reduced from 40% at 1 h after birth to 20% at 72 h. A study by Sezen et al. [23] found that the prevalence of RHs was only 2.6% after 3 days of birth. Unfortunately, other related factors that have been known to be associated with the occurrence of RHs, such as maternal age [11], primiparous mothers [23], duration and difficulty of active labor [21], converting delivery methods from NSVD to CS, and head circumference [24] were not assessed in this study. Future studies to specify more detailed risk factors in relation to the occurrence of RHs should be planned.

RHs in newborns have varying severity [2, 8, 10, 11]. A study by Watts et al. [8] reported that the severity of RHs ranged from mild (22–56%) to severe (18–37%) in their review article. A study by Emerson et al. [11] reported that RHs varied from a single dot hemorrhage in one eye to bilateral widespread hemorrhages. Furthermore, the characteristics of RHs vary according to their severity. RHs in newborns usually spontaneously resolve within 2 weeks, but in cases of severe RHs, it may persist for a longer duration and obscure the visual axis [3, 8, 9, 11,

12, 14]. This may limit the development of normal visual function, potentially resulting in complications such as anisometropia and amblyopia [14]. Thus, we divided newborns' eyes into the minimal, mild, moderate, and severe groups, and investigated the differences in characteristics according to the severity of RHs. Since no unified criteria are available for categorizing newborns and eyes with RHs, we used a newly developed criteria that further subdivided Watts et al.'s criteria [8]. This was possible because we could document the number of RHs more accurately by analyzing recorded retinal images obtained using WFDRI. Through a more detailed classification of the newborns' eyes, we could recognize the differences in characteristics according to the severity of RHs.

We found that bilateral RHs were more common than unilateral RHs in all the groups, as shown in previous studies [3, 5, 6, 8–11, 20]. In particular, most newborns had bilateral RHs (88.07%) in the severe group, but unilateral RHs were more common in the minimal group (56.51%). These results suggest that RHs in newborns may initially occur bilaterally due to the compression of the head during passage through the birth canal, and the RHs in one eye may disappear more quickly in the minimal group than in the severe group because of their small numbers. In the case of bilateral RHs, the severity in one eye usually matched that of the fellow eye (percentage of agreement = 71.4%; $P < 0.01$). These results are inconsistent with those of studies by Emerson et al. [11]. They reported that the severity of RHs in one eye did not correlate with the presence or severity of RHs in the fellow eye [11]. No laterality differences in RHs were noted in our study. This result is comparable with that of Simkin et al. [20] but is inconsistent with that of Callaway et al. [5], who reported that left-sided hemorrhages were far more common than were right-sided hemorrhages. These discrepancies may be due to the differences in the demographics of the enrolled newborns. In addition, the age at examination was significantly shorter in the severe group than in the other groups ($P < 0.001$). This result suggested that the age at examination affected not only the occurrence of RHs but also their severity.

Most RHs were intraretinal hemorrhages in all the groups (18678/21440; 87.12%). This is consistent with previously published data [8, 10, 11], but is inconsistent with that of Callaway et al. [5], who reported that 71% of RHs involved multiple layers of the retina. We hypothesized that intraretinal hemorrhages were predominant in all the groups because the retinal capillary plexus, which is located intraretinally, is more vulnerable to acute changes in arterial pressure than other vessels are because of its small size. In addition, approximately one-third of severe RHs showed preretinal hemorrhages in our study. This result suggested that the preretinally located retinal artery or arterioles, which are larger than the retinal capillary plexus, might be damaged because the newborns' heads would have been compressed by a stronger force during passage through the birth canal in the severe group than in the other groups.

RHs were mainly distributed in zone I in the minimal, mild, and moderate groups, and this result was comparable with that of Emerson et al. [11] RHs were mainly distributed in both zones I and II in the severe group, and RHs were rarely found in zone III. Chen et al. [3] found that all RHs were distributed in zone II, and Callaway et al. [5] reported that 95% of RHs involved the peripheral retina. These discrepancies may be due to the differences in the sample size and demographics of the enrolled newborns. Callaway et al. [5] also reported that RHs in newborns were most commonly ON flame hemorrhages (48.3%) and Roth spots (30.0%). Eighty-three percent of newborns with RHs had macular hemorrhages and 3.0% had foveal hemorrhages [5]. These characteristics seem to correspond to those of the severe group in our study. Most RHs did not involve the ON or macula, but both macular and ON involvement was high (7111/7355; 96.68%) in the severe group. Roth spots (7063/7355; 96.03%) were also commonly found in the severe group. Most of the RHs did not involve the fovea, but 374

(5.08%) eyes with severe RHs showed foveal involvement. These results suggest that newborns with RHs in the severe group had unique characteristics.

Notably, severe RHs in some newborns can be the result of nonaccidental trauma and Terson syndrome [13, 25–29] which is commonly associated with intracranial injury. As we have previously reported [13], RHs caused by abusive head trauma had a larger RH size, a higher percentage showed multilayer involvement and vitreous hemorrhage. Therefore, widespread severe RHs with multilayer involvement in the present study might have occurred in relation to the intracranial injuries. Unfortunately, we could not fully differentiate between RHs by birth canal compression and intracranial injury in the present study because of lack of information. Future studies about the pathophysiology of severe RHs should help provide an insight on this issue.

As mentioned above, the duration of RHs seems to depend on their severity, and severe RHs persist longer. We followed up newborns with severe RHs who visited our ophthalmology clinic and found that 126 (4.40%) eyes showed persistent RHs until 2 months after birth. Ma et al. [1] reported that all hemorrhages resolved spontaneously at the 3 months follow-up. Hughes et al. [10] found that 9 of 14 RHs resolved by 10 days, whereas dense foveal hemorrhages in 2 newborns persisted for a longer duration. Emerson et al. [11] reported that 85% of RHs resolved within 2 weeks, but a single subretinal hemorrhage persisted for longer than 4 weeks, and resolved only by 42 days.

In conclusion, this study confirmed that telemedicine combined with WFDRI was an effective and feasible method for investigating a large number of newborns with RHs. RHs commonly occur in healthy newborns and are significantly associated with NSVD. The characteristics of these newborns with RHs varied widely and seemed to depend on their severity. Severe RHs have unique characteristics, and future long-term longitudinal studies would be required to elucidate the prognosis of newborns with severe RHs.

## Author Contributions

**Conceptualization:** So Young Kim.

**Data curation:** In Hwan Cho, Nam Hun Heo, So Young Kim.

**Formal analysis:** Nam Hun Heo, So Young Kim.

**Funding acquisition:** So Young Kim.

**Investigation:** In Hwan Cho, So Young Kim.

**Methodology:** In Hwan Cho, Nam Hun Heo, So Young Kim.

**Validation:** In Hwan Cho, Min Seong Kim, So Young Kim.

**Writing – original draft:** In Hwan Cho, Min Seong Kim, So Young Kim.

**Writing – review & editing:** In Hwan Cho, So Young Kim.

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
