## [Decision Letter · Decision Letter 0]

6 Sep 2021

PONE-D-21-21462Birth-related retinal hemorrhages: The Soonchunhyang University Cheonan Hospital universal newborn eye screening (SUCH-NES) studyPLOS ONE

Dear Dr. Kim,

Thank you for submitting your manuscript to PLOS ONE. After careful consideration, we feel that it has merit but does not fully meet PLOS ONE’s publication criteria as it currently stands. Therefore, we invite you to submit a revised version of the manuscript that addresses the points raised during the review process.

ACADEMIC EDITOR: The manuscript is well-written and the number of included patients is quite impressive. I would suggest the authors highlight the difference between "clinical significance" and "statistical significance" in their study and discussion. The differences in gestational age and Apgar scores for example while might be statistically significant is too minute to be clinically significant.

We look forward to receiving your revised manuscript.

Kind regards,

Ahmed Awadein, MD, Ph.D, FRCS

Academic Editor

PLOS ONE

Reviewers' comments:

Reviewer's Responses to Questions

**Comments to the Author**

1. Is the manuscript technically sound, and do the data support the conclusions?

Reviewer #1: Yes

Reviewer #2: Yes

Reviewer #3: Yes

2. Has the statistical analysis been performed appropriately and rigorously? 

Reviewer #1: Yes

Reviewer #2: Yes

Reviewer #3: Yes

3. Have the authors made all data underlying the findings in their manuscript fully available?

Reviewer #1: Yes

Reviewer #2: Yes

Reviewer #3: Yes

4. Is the manuscript presented in an intelligible fashion and written in standard English?

Reviewer #1: Yes

Reviewer #2: Yes

Reviewer #3: Yes

5. Review Comments to the Author

Reviewer #1: Great paper. Congratulations.

This paper would be very helpful in enhancing our understanding of RH in neonates.

I have no concerns regarding ethics on this paper.

Well written paper that is easy to understand.

Reviewer #2: • This was a prospective, observational study aimed at evaluating the prevalence and clinical factors related to birth-related retinal hemorrhages. The large number of eyes is impressive, and makes up for the relatively short follow-up period.

• The authors regularly mention the term "civil clinics" which I am not familiar with, and could not find adequate information on. Please elaborate on what they are and whether they are different from regular clinics.

• Methods:

o The authors determined a history of systemic illness as an exclusion criterion. Does this include both maternal and fetal illness? Were there records on maternal blood disorders or medications that may influence clotting and predispose to hemorrhages?

o The authors do not mention any details on the length and difficulty of the vaginal deliveries. Were there appliances used in any of them? Please elaborate and compare this data to the described ocular findings.

o The authors do not mention whether any of the CS deliveries attempted vaginal deliveries first before converting to CS. This may skew the results in the CS group. Please confirm.

o The data suggests some of these newborns' APGAR scores were low. Were any of them distressed? Did any require oxygenation, intubation or any other medical procedures that may have contributed to the development of hemorrhages?

o The authors mention in page 6, line 121 that eyes with abnormal ocular findings were referred to a pediatric ophthalmologist. Were these eyes removed from the total cohort of eyes?

o The authors divided the eyes by severity into 4 groups based on the number of retinal hemorrhages. Please elaborate on the rationale behind the specific thresholds used in this classification.

o The authors mention that the images were reviewed by a single grader. Why wasn't a double-grader analysis used? Was the grader masked to the clinical history of the newborns they were assessing? Methodology involving subjective analysis of data usually involves double-grading and blinding to ensure the absence of bias and error. Please explain.

o The quality of some of the images used is questionable. One might argue that panel D2 in figure 1 is too hazy to determine the quantify and localize all hemorrhages in this eye. Please expand on this issue.

• Results:

o P. 7, lines 165-172: these numbers are already mentioned in table 2. There is no need to repeat them.

• Tables:

o Almost all tables require editing. The numbers and parentheses are not properly spaced.

• Minor edits:

o P.6, line 115: Add "s" to "hour".

Reviewer #3: This is the largest study to date of newborn retinal hemorrhages. The size is impressive. The main conclusion is that 20-25% of healthy newborns have some degree of retinal hemorrhages, the vast majority of which will resolve over then ensuing few weeks. As has been extensively described previously, NSVB is the main risk factor for retinal hemorrhages, theoretically due to compression though this has not been proven. Thus, while the main findings reported in this paper are not novel, they are certainly sound given the astronomical sample size. The authors should be commended for undertaking such a task.

My only concern is that some of the other associations reported seem to be so minuscule in terms of the absolute value of the difference that the clinical relevance is questionable. The association to female sex has a clinically irrelevant O.R. and the difference is a single percentage point. The difference in gestational age is less than 2 days. The difference in Apgar score at 1 minute is 0.04! These things are so minuscule that it makes me wonder about the distribution of the data and whether there is some batch effect leading to spurious findings. I would request that the authors provide some additional analysis to investigate possible confounders (perhaps redoing the statistics several times, each time with a single hospital removed) to see if the results are robust. I would also switch to non-parametric testing (e.g. rank-sum for continuous variables).

The discussion could benefit from more speculation about why these minuscule differences exist. There should also be more discussion about the full differential for retinal hemorrhages in a newborn beyond cranial compression (i.e. things like Terson's, clotting disorders, etc). In addition (perhaps most importantly), there is NO discussion about non-accidental trauma (and this CAN occur in the hospital). Figure 1-C1, Figure 2-D are very concerning and there should be some acknowledgement of this and discussion about any further exam or inquiry that was done given the appearance of widespread hemorrhage in several layers.

6. PLOS authors have the option to publish the peer review history of their article (what does this mean?). If published, this will include your full peer review and any attached files.

Reviewer #1: **Yes: **Donny Suh

Reviewer #2: No

Reviewer #3: No

---

## [Author Response · Author response to Decision Letter 0]

18 Oct 2021

October, 2021

The Editor,

PLOS ONE

Thank you for giving me the opportunity to submit a revised draft of my manuscript titled ‘Birth-related retinal hemorrhages: The Soonchunhyang University Cheonan Hospital universal newborn eye screening (SUCH-NES) study” to PLOS ONE. I appreciate the time and effort that you and the reviewers have dedicated in providing your valuable feedback on my manuscript. I am grateful to the reviewers for their insightful comments and suggestions which has helped me improve the quality of my submission. I have been able to incorporate changes to reflect most of the suggestions provided by the reviewers. Please note that I have highlighted the changes within the manuscript in red font.

Please find below the point-by-point response to the reviewers’ comments and concerns.

<Comments from Academic editor>

Comment : The manuscript is well-written and the number of included patients is quite impressive. I would suggest the authors highlight the difference between "clinical significance" and "statistical significance" in their study and discussion. The differences in gestational age and Apgar scores for example while might be statistically significant is too minute to be clinically significant.

Response : Thank you for your valuable comments. We completely agree with them. We think that gestational age and Apgar score showed statistical significance because the sample sizes were large. The magnitude of differences between groups might be too minimal to be clinically significant.

We have added the following sentences in the discussion section of our manuscript.

Modification : However, as the magnitude of differences of GA and Apgar scores at 1 and 5 min were minimal between the newborns with and without RHs, the clinical significance of these factors might be ambiguous (Page 25, Line 383-385).

<Comments from Reviewer #1>

Comment : Great paper. Congratulations. This paper would be very helpful in enhancing our understanding of RH in neonates. I have no concerns regarding ethics on this paper. Well written paper that is easy to understand.

Response : Thank you for your valuable comments and compliments.

<Comments from Reviewer #2>

Comment 1 : The authors regularly mention the term "civil clinics" which I am not familiar with, and could not find adequate information on. Please elaborate on what they are and whether they are different from regular clinics.

Response : Thank you for bringing this error to our notice. There was a mistake in the description of the term “civil clinics”. There are specialized obstetrics/gynecology (OB/GYN) hospitals in South Korea. These hospitals are responsible for the most of normal newborn deliveries. We collected data from these specialized OB/GYN hospitals via a telemedicine network. To reduce confusion, we used the term “OB/GYN hospitals” instead of “Civil OB/GYN clinics”.

Modification : Civil OB/GYN clinics -> OB/GYN hospitals

Please note that we have implemented this change throughout our revised manuscript.

Comment 2 : The authors determined a history of systemic illness as an exclusion criterion. Does this include both maternal and fetal illness? Were there records on maternal blood disorders or medications that may influence clotting and predispose to hemorrhages?

Response : Thank you for your pertinent observation. The history of systemic illness, both maternal and fetal, were collected. In addition, mothers with known blood disorders or medications that could influence the occurrence of retinal hemorrhages were excluded from our study. We have modified the exclusion criteria in the method section accordingly.

Modification : 

-> (2) maternal and fetal systemic diseases, (3) structural ocular anomalies, (4) a familial history of ocular congenital anomalies, and (5) those transferred to tertiary hospitals due to problems during delivery or (6) those undergoing the examination 7 days after birth were excluded from the study (Page 5, line 89-92).

Comment 3 : The authors do not mention any details on the length and difficulty of the vaginal deliveries. Were there appliances used in any of them? Please elaborate and compare this data to the described ocular findings.

Response : Thank you for your valuable observation. We agree with your opinion that there is a possibility that the duration and difficulty of the vaginal delivery can affect the occurrence of retinal hemorrhages. Unfortunately, we could not collect this data from each of the OB/GYN hospitals in the present study because of cost and labor restrictions. We are planning to analyze more details of risk factors in relation to the occurrence of retinal hemorrhages in a future study.

 In addition, these days, vacuum- or forceps- assisted vaginal delivery is not performed in South Korea. Relevant information regarding the same can be found in the discussion section and is as follows

 Unfortunately, since none of the newborns included in our study were delivered via vacuum- or forceps-assisted vaginal delivery, we could not elucidate the differences in the occurrence of RHs according to the mode of delivery. ( Page 25, line 378-381)

Modification : 

-> Unfortunately, other related factors that have been known to be associated with the occurrence of RHs, such as maternal age [11], primiparous mothers [23], duration and difficulty of active labor [21], converting delivery methods from NSVD to CS and head circumference [24], were not assessed in this study. Future studies to specify more detailed risk factors in relation to the occurrence of RHs should be planned (Page 25, line 395-400).

Comment 4 : The authors do not mention whether any of the CS deliveries attempted vaginal deliveries first before converting to CS. This may skew the results in the CS group. Please confirm.

Response : Thank you for the pertinent observation. Newborns with RHs in the CS group have a possibility that they were subject to attempted NSVD before converting to CS. Unfortunately, as we have mentioned above, these data could not be collected because of the cost and labor restrictions. We are planning to clarify these factors in a future study. 

Modification : 

-> Unfortunately, other related factors that have been known to be associated with the occurrence of RHs, such as maternal age [11], primiparous mothers [23], duration and difficulty of active labor [21], converting delivery methods from NSVD to CS and head circumference [24], were not assessed in this study. Future studies to specify more detailed risk factors in relation to the occurrence of RHs should be planned (Page 25, line 395-400).

Comment 5 : The data suggests some of these newborns' APGAR scores were low. Were any of them distressed? Did any require oxygenation, intubation or any other medical procedures that may have contributed to the development of hemorrhages?

Response : Thank you for your comments. Apgar score of 0 to 3 indicates a severely depressed neonate, whereas a score of 7 to 10 is considered normal. Here is the distribution of newborns according to the Apgar score at 1 min and 5 min from our study.

 Apgar score at 1 min Apgar score at 5 min

0 0 (0 %) 0 (0 %)

1 0 (0 %) 0 (0 %)

2 0 (0 %) 0 (0 %)

3 3 (0.01 %) 0 (0 %)

4 4 (0.01 %) 0 (0 %)

5 12 (0.02 %) 2 (0 %)

6 96 (0.17 %) 1 (0 %)

7 1176 (2.09 %) 71 (0.12 %)

8 9257 (16.49 %) 964 (1.71 %)

9 36696 (65.16 %) 9389 (16.61 %)

10 9003 (16.01 %) 45820 (81.46 %)

Total 56247 56247

As you commented, there were distressed newborns at 1 min after birth and the act of trying to revive them might have affected the occurrence of retinal hemorrhages. However, we think that the number of distressed newborns were too small to affect the main results of our study. In addition, there were no distressed newborns after 5 min of birth.

Comment 6 : The authors mention in page 6, line 121 that eyes with abnormal ocular findings were referred to a pediatric ophthalmologist. Were these eyes removed from the total cohort of eyes?

Response : Thank you for pointing this out. Abnormal ocular findings include all kinds of ocular abnormalities such as retinal hemorrhages, congenital cataract, optic disc abnormalities, retinoblastoma, and coloboma. Newborns with retinal hemorrhages without other ocular abnormalities were included our study, however, newborns with other ocular abnormalities were excluded from the study as we have mentioned in the exclusion criteria.

Comment 7 : The authors divided the eyes by severity into 4 groups based on the number of retinal hemorrhages. Please elaborate on the rationale behind the specific thresholds used in this classification.

Response : Thank you for pointing this out. We determined specific thresholds of the number of retinal hemorrhages from the study by Watts et al.[1] after modification.

 Since Watts et al. had determined the appropriate severity equivalent from previous numerous papers, [2-8] we think this can be a reasonable standard for dividing the groups. We have already commented about this in the discussion section. “Since no unified criteria are available for categorizing newborns and eyes with RHs, we used a newly developed criteria, that further subdivided Watts et al.’s criteria [8].”

 The modification can be also found in the method section.

Modification : Newborns and eyes with RHs were divided into four groups according to the number of RHs. -> Newborns and eyes with RHs were divided into four groups according to the number of RHs referring to the work of Watts et al. [8] after modification (Page 7, line 133-135). 

Comment 7 : The authors mention that the images were reviewed by a single grader. Why wasn't a double-grader analysis used? Was the grader masked to the clinical history of the newborns they were assessing? Methodology involving subjective analysis of data usually involves double-grading and blinding to ensure the absence of bias and error. Please explain.

Response : Thank you for your pertinent observation. In our telemedicine system, interpretation of images was done by a pediatric vitreoretinal specialist (SYK) who reported to the parents within 24 hours because of the cost and time constraints. However, during the preparation of the manuscript, we re-checked all the data by two independent graders (SYK and IHC) in a masked fashion. If there was any discrepancy between the graders, it was resolved by discussion. We apologize for missing this information in the method section.

 The modification can be found in the method section.

Modification : Retinal images from newborns with RHs were reviewed, and the details were recorded -> Retinal images from newborns with RHs were reviewed by two independent graders (SYK and IHC) in a masked fashion. If there was any discrepancy between the graders, consensus was reached by discussion (Page 7, line 128-130).

Comment 8 : The quality of some of the images used is questionable. One might argue that panel D2 in figure 1 is too hazy to determine the quantify and localize all hemorrhages in this eye. Please expand on this issue.

Response : Thank you for your valuable insights. As we have mentioned in the method section, if the captured images were of poor quality, the newborns were re-screened. However, in our humble opinion, we feel that the image quality of panel D2 in figure 1 depicts the necessary information. There were no major problems in the interpretation of the retinal hemorrhages according to the involved retinal layers. The image can be interpreted as below.

Comment 9 : P. 7, lines 165-172: these numbers are already mentioned in table 2. There is no need to repeat them.

Response : Thank you for your suggestion. We have removed the numbers which were already mentioned in table 2.

Modification : Among the screened newborns, newborns who met inclusion criteria were enrolled. In total, 56247 newborns from 61 OB/GYN hospitals were examined over 3 years and included 28894 males (51.37%) and 27353 females (48.63%). Newborns delivered via NSVD were 30826 (54.80%), and the rest were delivered via CS (25421/56247; 45.20%). The mean GA and mean BW were 273.10 ± 8.27 days and 3233.42 ± 405.13 g, respectively. The mean Apgar score was 8.95 ± 0.65 at 1 minute and 9.79 ± 0.46 at 5 minutes, and the mean age at examination was 2.00 ± 1.90 days (Table 2).

-> Among the screened newborns, newborns who met the inclusion criteria were enrolled. In total, 56247 newborns from 61 OB/GYN hospitals were examined over 3 years. Overall demographics of these newborns are described in Table 2 (Page 9, line 170-172).

Comment 10 : Tables: Almost all tables require editing. The numbers and parentheses are not properly spaced.

Response : Thank you for your feedback. Please note that we have edited and modified all the tables.

Comment 11 : Minor edits: P.6, line 115: Add "s" to "hour".

Response : Thank you for pointing out this error. We have made the necessary modifications.

Modification : hour -> hours (Page 6, line 116)

<Comments from Reviewer #3>

Comment 1 : This is the largest study to date of newborn retinal hemorrhages. The size is impressive. The main conclusion is that 20-25% of healthy newborns have some degree of retinal hemorrhages, the vast majority of which will resolve over then ensuing few weeks. As has been extensively described previously, NSVB is the main risk factor for retinal hemorrhages, theoretically due to compression though this has not been proven. Thus, while the main findings reported in this paper are not novel, they are certainly sound given the astronomical sample size. The authors should be commended for undertaking such a task

Response : Thank you for your valuable comments and compliments.

Comment 2 : My only concern is that some of the other associations reported seem to be so minuscule in terms of the absolute value of the difference that the clinical relevance is questionable. The association to female sex has a clinically irrelevant O.R. and the difference is a single percentage point. The difference in gestational age is less than 2 days. The difference in Apgar score at 1 minute is 0.04! These things are so minuscule that it makes me wonder about the distribution of the data and whether there is some batch effect leading to spurious findings. I would request that the authors provide some additional analysis to investigate possible confounders (perhaps redoing the statistics several times, each time with a single hospital removed) to see if the results are robust. I would also switch to non-parametric testing (e.g. rank-sum for continuous variables). The discussion could benefit from more speculation about why these minuscule differences exist.

Response : Thank you for your valuable comments. We completely agree with your comment. The female sex showed significant association with the occurrence of retinal hemorrhage in multivariate analyses. However, as mentioned in the discussion section, the odds ratio was slightly higher than 1, and the clinical significance of female sex might be ambiguous compared to that of male sex. You can find following sentence in the discussion section. “Female sex (OR, 1.050; 95% CI, 1.005–1.098; P = 0.029) was significantly associated with RHs. However, the OR was slightly higher than 1, and the clinical significance of female sex compared to male sex was ambiguous. Future studies are warranted to validate this result.”

 In case of gestational age and Apgar score, they were statistically significant because of large sample size. However, we think that the magnitude of differences between groups might be minimal to be clinically significant. We have added the following sentences in the discussion section of our manuscript. “But, because the magnitude of differences of gestational age and Apgar scores at 1 and 5 minutes were minimal between the newborns with and without RHs, clinical significance of these factors might be ambiguous.”

 As you pointed out a statistical problem with our manuscript, we have revisited our statistical analysis. Statistical analyses were conducted using both parametric and non-parametric statistical methods in total and by year (2017, 2018, and 2019). The gestational age was significantly different between groups in both parametric and non-parametric statistical methods from 2017 to 2019. The Apgar score at 1 min was significantly different between groups in both parametric and non-parametric statistical method in 2018 and 2019. From these results, there seems to be no major problem with our statistical analysis. Kindly let us know in case we are missing something. We will be happy to look into it again.

Comment 3 : There should also be more discussion about the full differential for retinal hemorrhages in a newborn beyond cranial compression (i.e. things like Terson's, clotting disorders, etc). In addition (perhaps most importantly), there is NO discussion about non-accidental trauma (and this CAN occur in the hospital). Figure 1-C1, Figure 2-D are very concerning and there should be some acknowledgement of this and discussion about any further exam or inquiry that was done given the appearance of widespread hemorrhage in several layers.

Response : Thank you for your valuable comments. We completely agree with your comments. Since fundus photography was optional, only newborns whose parents wanted the examination were screened. However, it might be difficult to exclude nonaccidental trauma. We have added comments about widespread severe RHs with multiple layer involvement which might be related to nonaccidental trauma or Terson syndrome in the discussion section. Unfortunately, it was impossible for us to fully differentiate between RHs by birth canal compression and intracranial injury in the present study because of lack of information. Since we are preparing future studies related to severe RHs, we expect to get an insight on this topic.

 The modification can be found in discussion section.

Modification : Notably, severe RHs in some newborns can be the result of nonaccidental trauma and Terson syndrome [13, 25-29] which is commonly associated with intracranial injury. As we have previously reported [13], RHs caused by abusive head trauma had a larger RH size, a higher percentage showed multilayer involvement and vitreous hemorrhage. Therefore, widespread severe RHs with multilayer involvement in the present study might have occurred in relation to the intracranial injuries. Unfortunately, we could not fully differentiate between RHs by birth canal compression and intracranial injury in the present study because of lack of information. Future studies about the pathophysiology of severe RHs should help provide an insight on this issue. (Page 28, line 457-465).

References

1. Watts P, Maguire S, Kwok T, Talabani B, Mann M, Wiener J, et al. Newborn retinal hemorrhages: a systematic review. J AAPOS. 2013;17(1):70-8. Epub 2013/02/01. doi: 10.1016/j.jaapos.2012.07.012. PubMed PMID: 23363882.

2. Berkus MD, Ramamurthy RS, O'connor PS, Brown K, Hayashi RHJO, gynecology. Cohort study of silastic obstetric vacuum cup deliveries: I. Safety of the instrument. 1985;66(4):503-9.

3. Emerson MV, Pieramici DJ, Stoessel KM, Berreen JP, Gariano RF. Incidence and rate of disappearance of retinal hemorrhage in newborns. Ophthalmology. 2001;108(1):36-9. Epub 2001/01/11. doi: 10.1016/s0161-6420(00)00474-7. PubMed PMID: 11150261.

4. Hughes LA, May K, Talbot JF, Parsons MA. Incidence, distribution, and duration of birth-related retinal hemorrhages: a prospective study. J AAPOS. 2006;10(2):102-6. Epub 2006/05/09. doi: 10.1016/j.jaapos.2005.12.005. PubMed PMID: 16678742.

5. Kuit JA, Eppinga HG, Wallenburg H, Huikeshoven FJO, gynecology. A randomized comparison of vacuum extraction delivery with a rigid and a pliable cup. 1993;82(2):280-4.

6. Schoenfeld A, Buckman G, Nissenkorn I, Cohen S, Ben-Sira I, Ovadia JJAoO. Retinal hemorrhages in the newborn following labor induced by oxytocin or dinoprostone. 1985;103(7):932-4.

7. Svenningsen L, Eidal KJAoegS. Lack of correlation between umbilical artery pH, retinal hemorrhages and Apgar score in the newborn. 1987;66(7):639-42.

8. Williams MC, Knuppel RA, O'Brien WF, Weiss A, Spellacy WN, Pietrantoni M. Obstetric correlates of neonatal retinal hemorrhage. Obstet Gynecol. 1993;81(5 ( Pt 1)):688-94. Epub 1993/05/01. PubMed PMID: 8469455.

---

## [Editor Report · Decision Letter 1]

19 Oct 2021

Birth-related retinal hemorrhages: The Soonchunhyang University Cheonan Hospital universal newborn eye screening (SUCH-NES) study

PONE-D-21-21462R1

Dear Dr. Kim,

We’re pleased to inform you that your manuscript has been judged scientifically suitable for publication and will be formally accepted for publication once it meets all outstanding technical requirements.

Kind regards,

Ahmed Awadein, MD, Ph.D, FRCS

Academic Editor

PLOS ONE
---

## [Editor Report · Acceptance letter]

29 Oct 2021

PONE-D-21-21462R1 

Birth-related retinal hemorrhages: The Soonchunhyang University Cheonan Hospital universal newborn eye screening (SUCH-NES) study 

Dear Dr. Kim:

I'm pleased to inform you that your manuscript has been deemed suitable for publication in PLOS ONE. Congratulations! Your manuscript is now with our production department. 

Kind regards, 

on behalf of

Dr. Ahmed Awadein 

Academic Editor

PLOS ONE